# A Low-Intensity Internet-Based Intervention Focused on the Promotion of Positive Affect for the Treatment of Depression in Spanish Primary Care: Secondary Analysis of a Randomized Controlled Trial

**DOI:** 10.3390/ijerph17218094

**Published:** 2020-11-03

**Authors:** Mª Dolores Vara, Adriana Mira, Marta Miragall, Azucena García-Palacios, Cristina Botella, Margalida Gili, Pau Riera-Serra, Javier García-Campayo, Fermín Mayoral-Cleries, Rosa Mª Baños

**Affiliations:** 1Polibienestar Research Institute, University of Valencia, 46022 Valencia, Spain; banos@uv.es; 2CIBERObn Physiopathology of Obesity and Nutrition, Instituto de Salud Carlos III, 28029 Madrid, Spain; marta.miragall@uv.es (M.M.); azucena@uji.es (A.G.-P.); botella@uji.es (C.B.); 3Department of Personality, Evaluation and Psychological Treatment, Faculty of Psychology, University of Valencia, 46010 Valencia, Spain; adriana.mira@uv.es; 4Department of Basic and Clinical Psychology and Psychobiology, Faculty of Health Sciences, Jaume I University, 12071 Castellon de la Plana, Spain; 5Institut Universitari d’Investigació en Ciències de la Salut, University of Balearic Islands, E-07122 Palma de Mallorca, Spain; mgili@uib.es (M.G.); pau.riera@uib.es (P.R.-S.); 6Institut d’Investigació Sanitaria Illes Balears, 07120 Palma de Mallorca, Spain; 7Primary Care Prevention and Health Promotion Research Network, RedIAPP, 28029 Madrid, Spain; jgarcamp@unizar.es; 8Aragon Institute for Health Research (IIS Aragón), Miguel Servet Hospital, University of Zaragoza, 50009 Zaragoza, Spain; 9Mental Health Unit, Hospital Regional of Malaga, Biomedicine Research Institute (IBIMA), 29010 Málaga, Spain; fermin.mayoral.sspa@juntadeandalucia.es

**Keywords:** depression, primary care, internet-based intervention, positive affect

## Abstract

*Background*: A large number of low-intensity Internet-based interventions (IBIs) for the treatment of depression have emerged in Primary Care; most of them focused on decreasing negative emotions. However, recent studies have highlighted the importance of addressing positive affect (PA) as well. This study is a secondary analysis of a randomized control trial. We examine the role of an IBI focused on promoting PA in patients with depression in Primary Care (PC). The specific objectives were to explore the profile of the patients who benefit the most and to analyze the change mechanisms that predict a significantly greater improvement in positive functioning measures. *Methods*: 56 patients were included. Measures of depression, affect, well-being, health-related quality of life, and health status were administered. *Results*: Participants who benefited the most were those who had lower incomes and education levels and worse mental health scores and well-being at baseline (7.9%–39.5% of explained variance). Improvements in depression severity and PA were significant predictors of long-term change in well-being, *F* (3,55) = 17.78, *p* < 0.001, R^2^ = 47.8%. *Conclusions*: This study highlights the importance of implementing IBIs in PC and the relevance of PA as a key target in Major Depressive Disorder treatment.

## 1. Introduction

Major depressive disorder (MDD) is one of the most prevalent mental disorders worldwide, and it is associated with significant personal and social costs [1,2]. Specifically, the prevalence of depression in Spanish Primary Care (PC) is between 13.9 and 29% [3]. Even though there are effective treatments for this disorder (pharmacotherapy, psychotherapy, or both) [4,5], only half the people with depression receive adequate care [6], only two-thirds of patients respond to treatment, and only about one-third experience remission of their depressive symptoms [7,8]. Therefore, it is important to continue to explore different ways to improve these treatment outcomes.

Currently, there are evidence-based psychological treatments (e.g., cognitive-behavioral therapy, interpersonal therapy, problem-solving therapy) for depressive disorders in PC [9]. However, their therapeutic approaches have focused mainly on improving negative affect (NA) (e.g., depression, anxiety), paying less attention to the role of promoting positive affect (PA) and building positive resources (e.g., well-being, strengths) [10,11]. The literature indicates that MDD is related to low levels of positive emotions and that depression is more associated with low levels of PA than other emotional disorders [12]. Several studies point out the importance of promoting PA—as well as gratitude, resilience, and positive functioning—as core elements of interventions to facilitate recovery from depression [13,14].

A possible alternative in understanding MDD is to consider it a heterogeneous diagnostic construct with different underlying functional domains that may require different therapeutic strategies, as proposed in the Research Domain Criteria (RDoC) and transdiagnostic approaches [15]. Within this framework, MDD is proposed as resulting from alterations in two partly dissociable neurobiological dimensions: the upregulation of a negative valence system that promotes NA and leads to a pervasive depressed mood; and the downregulation of a positive valence system that guides the focus toward rewarding stimuli or PA, leading to loss of interest or pleasure (anhedonia) [16,17]. In the same vein, to effectively treat depression, research needs to propose psychotherapeutic interventions that include elements to downregulate NA and upregulate PA.

Several therapeutic approaches from the positive psychology (PP) field [18] are emerging with the aim of enhancing positive emotional human functioning (e.g., well-being, resilience, satisfaction with life), especially PA. Some reviews and meta-analyses show that PP interventions are effective in reducing depressive symptoms and increasing well-being [19,20,21,22], emphasizing their role alongside standard treatments for clinical depression [23].

One of the most important obstacles to integrating these treatments in PC settings is the lack of time and resources [24]. Many of the empirically validated treatments for MDD consist of 15–20 sessions requiring one hour per week [25,26]. Although this treatment length is considered quite brief compared to previous approaches, it has been argued that it is still too intense to be implemented in PC. One way to reduce the high costs of MDD treatment and overcome the limitations of traditional treatments in PC (e.g., logistical, geographical, and access difficulties) is to propose the use of brief or low-intensity psychological interventions for the treatment of MDD. Internet-based interventions (IBIs) are an example of low-intensity interventions that have been shown to be appropriate and cost-effective options within the stepped care model for the treatment of MDD [27]. Given the potential of IBIs as low-intensity interventions in PC, we conducted a randomized control trial (RCT) in a sample of patients diagnosed with MDD in different PC settings in Spain [28]. In this study, we examined the effectiveness of three low-intensity IBIs (PA promotion vs. healthy lifestyle vs. mindfulness) along with improved treatment as usual (iTAU), compared to iTAU alone, and the results were promising [29]. Specifically, healthy lifestyle and mindfulness interventions were effective in reducing depression severity, compared to iTAU, at post-treatment. Moreover, all the interventions were also effective in improving medium- and long-term quality of life. Finally, the PA promotion intervention was found to be effective in improving well-being at the six-month follow-up and NA at the 12-month follow-up, compared to iTAU.

The present study is a secondary analysis of the RCT carried out by Gili et al. [29]. More specifically, this study carries out an in-depth examination of the role of a specific IBI to promote PA as an adjunct therapy to iTAU in PC settings, given PA’s potential as a strategy to improve well-being in MDD patients. Hence, the specific objectives are to determine: (1) which patient profile (i.e., sociodemographic and clinical baseline variables) predicts the improvement in depression severity and positive functioning in the post-treatment and follow-up sessions (Objective 1); and (2) which change mechanism predicts the greatest change in measures related to positive functioning in the post-treatment and follow-up sessions (Objective 2). The results on the efficacy of the PA intervention (vs. iTAU condition) can be found in Gili et al. [29]. Nevertheless, we will also analyze the changes in the primary and secondary outcomes from pre-treatment to post-treatment and at the 6- and 12-month follow-ups to provide a better understanding of the results of Objective 1 and 2. No specific hypotheses are proposed due to the exploratory nature of the analyses.

## 2. Materials and Methods

### 2.1. Study Design 

This is a secondary analysis study of an RCT with repeated measures (baseline, post-treatment, 6-month and 12-month follow-ups) with four independent conditions: (a) PA promotion IBI + iTAU; (b) healthy lifestyle psychoeducational IBI + iTAU; (c) mindfulness IBI + iTAU; and (d) only iTAU group, in PC ([29]; Current Controlled Trials ISRCTN82388279). In the present study, we will only focus on the condition of the PA promotion IBI + iTAU.

### 2.2. Participants

In the original study, a total of 221 participants were recruited in PC settings from the Spanish regions of Aragon, Andalusia, and the Balearic Islands. The present study included 56 participants (78.6% women) who completed the PA intervention. Ages ranged between 19 and 68 years, with a mean of 44.14 years (*SD* = 10.38). In addition, 55.4% of the participants were married, 80.4% lived with family or a partner, 28.6% had higher education, 44.6% were employed, and 23.2% had an income below the national minimum wage. Regarding depression severity at baseline, the average on the Spanish Patient Health Questionnaire-9 (PHQ-9) was 15.79 (*SD* = 6.21). 

Inclusion criteria were: (a) having mild or moderately severe depressive symptoms according to the Spanish Patient Health Questionnaire-9 (PHQ-9; [30]) (5–9 = Mild depression; 10–14 = moderate depression); (b) age between 18–65 years; (c) ability to use a computer; (d) having Internet and an email account; and (e) being able to read and understand Spanish. The Mini International Neuropsychiatric Interview (MINI) 5.0 [31,32] was used to assess different mental disorders and establish the diagnosis. Patients were excluded from the study if they had (a) severe depression (score ≥14 on the PHQ-9); (b) a severe Axis I psychiatric disorder (e.g., psychotic disorders, presence of suicidal ideation or plan, alcohol/substance abuse or dependence); (c) any disease that can affect the central nervous system (e.g., brain pathology, traumatic brain injury, dementia); or (d) if they were currently receiving psychological treatment.

Full information on the participant flow of the PA intervention is shown in Figure 1.

### 2.3. Interventions

All the participants in the PA condition completed the low-intensity IBI and received iTAU from their general practitioner (GP).

#### 2.3.1. Low-Intensity Internet-Based Computerized Intervention (IBI) Focused on the Promotion of PA

Before starting the online treatment, participants received a 90-min (3–5 patients) group face-to-face session conducted by a clinical psychologist. The objective of this session was to explain the program structure and the main components of the treatment, clarify the instructions for the use of the online platform, and motivate participants to change.

The online self-guided program consisted of four therapeutic modules (60 min per module approximately). The program duration could vary among the users, but it was usually completed in 4 to 8 weeks (maximum of 2 weeks per module). These modules contained multimedia elements (videos, images, texts) that provided information about MDD and coping strategies. These modules were sequential so that users could move step by step through the program. Moreover, users could review the module’s contents after they had finished. 

To enhance adherence, participants received two weekly automated mobile phone messages encouraging them to proceed with the program and reminding them of the importance of doing the homework tasks. In addition, the participants received automated emails encouraging them to continue with the modules if they had not accessed the program for a week. 

Specifically, this intervention was based on PP techniques, and it was mainly designed to decrease depression severity and prevent relapse by promoting PA and subjective well-being. Table 1 shows each module, its objectives, and its specific content. For more information about the protocol for promoting PA, see García-Palacios, Mira, Mayoral, Baños, and Botella [33].

#### 2.3.2. iTAU

The Treatment as Usual (TAU) in PC settings was improved. The participating GP received a training program on how to diagnose and treat depression in PC, based on the Spanish Guide for the Treatment of Depression in Primary Care [34]. Participants could receive medication for at least two months, and they had at least four visits with the GP that lasted an average of 30 min each. In cases with suicide risk, severe social dysfunction, or worsening of symptoms, the GP had to refer the patient to mental health facilities. 

### 2.4. Measures 

#### 2.4.1. Primary Outcome Measure

*Depression severity:* The Patient Health Questionnaire-9 (PHQ-9; [35]) is a 9-item self-administered test for screening, diagnosing, monitoring, and measuring depression severity. Patients describe their state, taking into account the two weeks before the evaluation. Items are rated from 0 to 3, denoting “not at all”, “several days”, “more than half the days”, and “nearly every day”, respectively. Cut-off points of 5, 10, 15, and 20 represent mild, moderate, moderately severe, and severe depression (DSM-IV-TR; [36]). The Spanish version has been shown to have good psychometric properties (for the diagnosis of any disorder, k = 0.74; overall accuracy, 88%; sensitivity, 87%; specificity, 88%) [30]. In the current study, the alpha coefficient was very satisfactory (α = 0.86).

#### 2.4.2. Secondary Outcomes Measures 

*Affect:* The Positive and Negative Affect Scale (PANAS; [37]) consists of 20 items that evaluate two independent dimensions: PA (10 items) and NA (10 items). The total score for each subscale ranges from 5 to 50, using a 5-point Likert-type scale (1 = very slightly or not at all, 5 = very much). The Spanish version by Sandín et al. [38] showed adequate internal consistency. In the current study, alpha coefficients were very satisfactory for both scales (αs = 0.87 and 0.86 for positive and negative subscales, respectively).

*Well-being:* The Pemberton Happiness Index (PHI; [39]) consists of 11 items related to different domains of remembered well-being (general, hedonic, eudaimonic, and social well-being), each with an 11-point Likert-type scale (0 = strongly disagree, 10 = strongly agree), and 10 items related to experienced well-being (positive and negative emotional events that may have happened the day before), with dichotomous response options (yes/no). The sum of the remembered and experienced well-being scores yields a combined well-being index (total well-being) ranging from 0 to 10. The validated Spanish version showed adequate psychometric properties [39]. In the current study, the alpha coefficient was very satisfactory (α = 0.86). In the current study, the alpha coefficient was very satisfactory (αs = 0.80).

*Health-related quality of life:* The Short Form 12 Health Survey (SF-12; [40]) is a 12-item questionnaire that measures aspects of health-related quality of life from the patient’s perspective in two dimensions: physical health (general health, bodily pain, role-physical, physical functioning) and mental health (mental health, role-emotional, social-functioning, vitality). Scores > 50 represent better physical or mental health than the mean, and scores <50 represent worse physical or mental health than the mean. The Spanish version has been found to be a valid and reliable measure, showing good internal consistency [41]. In the current study, alpha coefficients were acceptable for both scales (αs = 0.80 for the physical health dimension and 0.66 for the mental health dimension, respectively).

*Health status:* The EuroQoL (EQ-5D; [42]) is a 5-item instrument to assess general health status (mobility, self-care, pain, usual activities, and psychological status) with three possible answers for each item (1 = no problem, 2 = moderate problem, 3 = severe problem). A summary index with a maximum score of 1 can be derived from these five dimensions using a conversion table [42]. The maximum score of 1 indicates the best health state, in contrast to the scores on individual questions, where higher scores indicate more severe or frequent problems. In addition to the descriptive system, the EQ-5D includes a visual analog scale (EQ VAS). The EQ VAS is a rating scale ranging from 0 (worst imaginable health state) to 100 (best imaginable health state), and it represents the evaluation of the patient’s health state from his/her point of view. The validated Spanish version of the EQ-5D was used [42]. In this study, only the EQ VAS was taken into account.

#### 2.4.3. Screening Related Measures

*Sociodemographic data:* Personal data that includes information such as age, sex, marital status, living alone or with others, educational level, employment, and income level.

*Diagnostic interview:* The MINI International Neuropsychiatric Interview version 5.0 (M.I.N.I. 5.0; [31]) is used in the screening to assess current depression and comorbid disorders. This measure is a structured diagnostic interview based on the DSM-IV and ICD-10 criteria. The Spanish version [32] was used for this study.

### 2.5. Procedure

Patients were recruited in Spanish PC settings from three different regions between March 2015 and March 2016. GPs detected possible participants with MDD using the PHQ-9. After a few days, an independent researcher used the MINI to assess the participants, taking into account the inclusion and exclusion criteria. Participants gave their signed written informed consent to be part of the study. Then, randomization was carried out by another independent researcher. All participants completed the pre-treatment assessment integrated into a web system (https://psicologiaytecnologia.labpsitec.es/). At the end of the treatment, they also completed the post-treatment and follow-up assessments through the website. The PHQ-9, PANAS, and SF-12 were administered at baseline, post-treatment, and 6-month and 12-month follow-ups. The PHI and EQ-5D were administered only at baseline and in the follow-up sessions. The Ethical Review Board of the regional health authority approved the study (Ref: IB 2144/13PI). More details about the study design, recruitment, and randomization methods are included in the protocol study by Castro et al. [28].

### 2.6. Data Analyses

All statistical analyses were performed using the SPSS v.26 (IBM Corp, Armonk, NY, USA). First, we implemented Multiple Imputation with Chained Equations (MICE) to replace the outcomes’ missing values, performing 100 imputation models with 100 iterations per model [43].

Second, preliminary analyses were conducted to ensure that relevant assumptions of repeated-measures ANOVAs and multiple regression were met (i.e., normality, sphericity, and absence of multicollinearity). We tested normality, carrying out a visual inspection of the Q-Q (quantile-quantile) plots, verifying that the observed data was approximately closed to the expected data (i.e., the distance between the observed and expected data was not extreme). Hence, we assured that we met the normality assumption to carry out a repeated-measures ANOVA and multiple regression. Moreover, we tested the assumption of sphericity using Mauchly’s test. If Mauchly’s test statistic was significant (i.e., sphericity was not met), the degrees of freedom were adjusted using the Greenhouse–Geisser correction. Finally, we tested the absence of multicollinearity using the Variance Inflation Factor (VIF).

Third, seven repeated-measures ANOVA with time as within-factor—pre-treatment (Pre), post-treatment (Post), 6-month follow-up (FW6), and 12-month follow up (FW12)—were conducted to analyze the changes in each primary (i.e., PHQ-9) and secondary (i.e., PANAS-PA, PANAS-NA, PHI, SF-12 mental and physical health, EQ-5D) outcomes. Post-hoc analyses using Bonferroni corrections were carried out when significant effects were found. Within-group Cohen’s d effect sizes with a 95% Confidence Interval were calculated. 

Fourth, seven hierarchical multiple regression analyses with a stepwise selection of predictors within each block were conducted to explore which sociodemographic variables (i.e., sex, age, marital status, living alone or with others, educational level, employment, and income level) and pre-treatment scores on each primary and secondary outcome predicted the pre-post treatment, pre-FW6, and pre-FW12 change. The seven sociodemographic variables were entered in the first block, and six pre-treatment scores were entered in the second block to test the relevance of the extra explained variance of these variables in the dependent variables once the effects of the sociodemographic variables were controlled for. Given the small sample size to the number of predictors, a consideration of all predictors simultaneously in each regression was not tenable. Therefore, within each block, a statistical inclusion criterion for relevant predictors (stepwise method) was used. Creating two blocks in the regression analyses allowed us to test a lower number of predictors in each regression (i.e., seven instead of thirteen predictors). 

To carry out correlation and regression analyses, categorical sociodemographic variables were transformed into recoded binary variables; that is, the correlation and regression analyses included: dichotomous variables (i.e., sex), continuous variables (i.e., age and scores in PHQ-9, PANAS-PA, PANAS-NA, PHI, SF-12 mental and physical health, EQ-5D) and recoded binary variables (i.e., ordinal or categorical variables were recoded into dichotomous variables with 0 and 1). Regarding the recoded binary variables, the categories were recategorized as follows: (a) marital status: 0 = not married (i.e., single, divorced, widowed); 1 = married or in a relationship; (b) *living alone or with others:* 0 = living alone; 1 = living with others (i.e., partner, sons, relatives, friends); (c) *educational level:* 0 *=* lower level (i.e., no education or primary school); 1 = higher level (i.e., secondary school or university studies); (d) *work status:* 0 *=* “not-working” (i.e., unemployed, retired, housekeeper, disability, student); 1 = employed; (e) *income level*: 0 = “lower than the minimum income (i.e., <641.40€); 1 = higher than the minimum income. We recoded into binary variables (instead of doing dummy variables) because the pair comparisons we constructed were theoretically appropriate, and a smaller number of predictors should be tested in the regression analyses. That is, if binary variables are used, only one predictor should be tested (e.g., 0 = “lower than the minimum income (i.e., <641.40€) vs. 1 = higher than the minimum income); however, if dummy variables are used, more than one predictor should be included in the regression equation (e.g., *dummy 1*: 0 = “lower than the minimum income < 641.40€ vs. 1 = between 1–2 minimum incomes; *dummy 2*: 0 = “lower than the minimum income < 641.40€ vs. 1 = between 2–4 minimum incomes). Moreover, the change in each primary and secondary outcome was calculated as follows: pre-post (post-treatment scores—pre-treatment scores), pre-FW6 (6-month follow up—pre-treatment scores), and pre-FW12 (12-month follow-up—pre-treatment scores). Negative values for the changes in PHQ and PANAS-NA meant improvements in these measures, whereas negative values for the changes in PANAS-PA, PHI, SF-12, and 5Q-5D meant a deterioration in these measures. The pairwise deletion method was used (i.e., whenever the variables of interest are present, they are analyzed) to deal with missing values and preserve all the data available in the regression analyses. The pre-treatment score on the corresponding dependent variable was not introduced in the equation regression model (e.g., the PHQ-9 pre score was not introduced as a predictor when the change in PHQ-9 was tested as a dependent variable). 

Finally, four stepwise multiple regression analyses were conducted to analyze whether the changes in PHQ, PANAS-PA, or PANAS-NA were predictors of the change in PHI, SF-12 (mental and physical health), and EQ-5D. Changes in pre-post predictors were introduced for changes in pre-post dependent variables; changes in pre-post and pre-FW6 predictors were introduced for changes in pre-FW6 dependent variables; and changes in pre-post, pre-FW6, and pre-FW-12 predictors were introduced for changes in pre-FW12 dependent variables. All predictor variables were entered in the same block using the stepwise method.

It should be noted that the stepwise approach was used because we did not have an a priori hypotheses of which specific independent variables would predict the dependent variables, and consequently, we decided to identify the predictor variables relying on a statistical criterion.

## 3. Results

### 3.1. Sociodemographic Characteristics of the Sample

Table 2 shows a detailed description of the sociodemographic variables.

### 3.2. Primary and Secondary Outcome Scores at Pre-Treatment, Post-Treatment, and 6- and 12-Month Follow-Ups

Descriptive statistics, repeated-measures ANOVA results, and within-group effect sizes with 95% CI are shown in Table 3 for primary and secondary outcomes. The main effects of time were found for all the primary and secondary outcomes, with increases in the PANAS-PA, PHI, SF-12, and EQ-5D scores and decreases in the PHQ-9 and PANAS-NA scores.

Post-hoc analyses using the Bonferroni correction showed that there were significant differences from pre-treatment to post-treatment, FW6, and FW12 in depression severity (PHQ-9), PA (PANAS-PA), NA (PANAS-NA), and Mental Health (SF-12) (*p* < 0.05). Significant differences from pre-treatment to FW12 were found for Physical Health (SF-12), but not from pre-treatment to post-treatment or FW6. Finally, significant differences from baseline to the 6- and 12-month follow-ups were found for well-being (PHI) and health status (EQ-5D).

### 3.3. Sociodemographic Variables and Pre-Treatment Scores as Predictors of the Changes in Primary and Secondary Outcomes in the PA Intervention

Point-Biserial and Pearson’s correlations between potential predictors (sociodemographic and pre-treatment scores) are shown in Table 4. Positive significant relationships were found between age and educational level and income level; being married or in a relationship correlated positively with well-being (PHI) and with living with someone, and income level correlated positively with health status (5Q-5D). A negative significant relationship was found between age and physical health (SF-12). Negative significant relationships were found between depression severity (PHQ-9) and positive affect (PANAS), well-being (PHI), mental health and physical health (SF-12), and health status (5Q-5D). Finally, a positive significant relationship was found between depression severity (PHQ-9) and NA (PANAS). 

Regarding hierarchical multiple regression analysis, coefficients of determination, unstandardized coefficients, standard errors, standard coefficients, and *t*-statistics are shown in Table 5. The statistical models for predicting the change in each primary and secondary outcome are described in the following paragraphs. Variance Inflation Factor ranged from 0.95 to 1.06 for all the regression analyses, indicating no problems with multicollinearity [44,45].

*Models for predicting change in depression severity (PHQ).* Level of income positively predicted the change from Pre-Post, *F*(1,38) = 10.35, *p* = 0.003, R^2^ Adjusted = 19.7%; Level of income positively predicted the change from Pre-FW6, *F*(1,38) = 5.50, *p* = 0.024, R^2^ Adjusted = 10.6%, and level of income positively predicted the change from Pre-FW12, *F*(1,38) = 16.00, *p* < 001, R^2^ Adjusted = 28.3%. Once the sociodemographic variables were controlled, only well-being (PHI) positively predicted the change in PHQ from Pre-FW12, *F*(2,38) = 13.39, *p* < 0.001, R^2^ Adjusted = 39.5%, which significantly increased the explained variance in 12.5% (*p* = 0.008). Lower level incomes predicted greater improvements in depression severity at post-treatment, and the 6- and 12-month follow-ups; and lower well-being predicted greater improvements in depression severity at the 12-month follow-up.

*Models for predicting change in PA (PANAS).* None of the variables entered predicted the Pre-Post, Pre-FW6, or Pre-FW12 change.

*Models for predicting change in NA (PANAS).* None of the variables entered predicted the Pre-Post, Pre-FW6, or Pre-FW12 change.

*Models for predicting change in well-being (PHI).* Mental health (SF-12) negatively predicted the change Pre-FW6 change, *F*(1,38) = 4.27, *p* = 0.046, R^2^ Adjusted = 7.9%. None of the variables entered predicted the Pre-FW12 change. Lower mental health predicted greater improvements in well-being at the 6-month follow-up.

*Models for predicting change in mental health (SF-12).* Level of income negatively predicted the Pre-FW6 change, *F*(1,38) = 5.71, *p* = 0.022, R^2^ Adjusted = 11.0%. None of the variables entered predicted the Pre-FW12 change. Lower income levels predicted greater improvements in mental health at the 6-month follow-up.

*Models for predicting change in physical health (SF-12).* Educational level predicted negatively the change from Pre-Post, *F*(1,38) = 4.33, *p* = 0.044, R^2^ Adjusted = 8.1%; Pre-FW6, *F*(1,38) = 8.23, *p* = 0.007, R^2^ Adjusted = 16.0%, and Pre-FW12, *F*(1,38) = 9.04, *p* = 0.005, R^2^ Adjusted = 17.5%. Lower levels of education predicted greater improvements in physical health at post-treatment and 6- and 12-month follow-ups.

*Models for predicting change in health status (EQ-5D).* Level of income negatively predicted the Pre-FW6 change, *F*(1,38) = 10.20, *p* = 0.003, R^2^ Adjusted = 19.5%. None of the variables entered predicted the Pre-FW12 change. Lower income levels predicted greater improvements in health status at the 6-month follow-up.

### 3.4. Change in Depression Severity and PA and NA as Predictors of Change in Well-Being, Mental and Physical Health, and Health Status

Coefficients of determination, unstandardized coefficients, standard errors, standard coefficients, and *t*-statistics for each stepwise multiple regression analysis are shown in Table 6. The statistical models to explain the changes in well-being, mental and physical health and health status are described in the following paragraphs. Variance Inflation Factor ranged from 1.01 to 2.37 for all the regression analyses, indicating no problems with multicollinearity.

*Models for predicting change in well-being (PHI).* The Pre–Post change in depression severity (PHQ-9) negatively predicted the Pre-FW6 change in well-being, *F*(1,55) = 6.60, *p* = 0.013, R^2^ Adjusted = 9.2%. The Pre-12FW change in depression severity (PHQ-9) and PA (PANAS-PA), and the Pre-6FW change in depression severity (PHQ-9) (negatively, positively and negatively -respectively-) predicted the Pre-FW12 change in well-being, *F*(3,55) = 17.78, *p* < 0.001, R^2^ Adjusted = 47.8%. Greater improvements in depression severity at post-treatment predicted greater improvements in well-being at the 6-month follow-up and greater improvements in depression severity and PA at the 12-month follow-up, and greater improvements in depression severity at the 6-month follow-up predicted greater improvements in well-being at the 12-month follow-up.

*Models for predicting change in mental health (SF-12).* The Pre-FW6 change in depression severity (PHQ-9) negatively predicted the Pre-FW6 change in mental health, *F*(1,55) = 7.37, *p* = 0.009, R^2^ Adjusted = 10.4%. Greater improvements in depression severity at the 6-month follow-up predicted greater improvements in mental health at the 6-month follow-up.

*Models for predicting change in physical health (SF-12).* Changes in Pre-Post treatment scores for depression severity (PHQ-9) and NA (PANAS-NA) negatively predicted the Pre-Post change in physical health, *F*(1,55) = 5.19, *p* = 0.009, R^2^ Adjusted = 13.2%. Greater improvements in depression severity and NA at post-treatment predicted greater improvements in physical health at post-treatment.

*Models for predicting change in health status* (EQ-5D). The Pre-FW12 change in depression severity (PHQ-9) negatively predicted the Pre-FW12 change in health status, *F*(1,55) = 5.91 *p* = 0.018, R^2^ Adjusted = 8.2%. Greater improvements in depression severity at the 12-month follow-up predicted greater improvements in health status at the 12-month follow-up.

## 4. Discussion

The purpose of this study was to shed light on the *patient profiles* (Objective 1) and *change mechanisms* (e.g., PA or NA) (Objective 2) that predicted a significantly greater improvement in depressive severity and positive functioning variables in an IBI focused on promoting PA, along with iTAU, in patients with MDD.

The efficacy of this intervention was tested in Gili et al. [29], where the PA intervention is compared to other interventions and a control group. However, in this study, we analyzed the change in the primary and secondary treatment outcomes to contextualize the secondary analyses of the present study. The results showed that participants improved on all the variables evaluated from pre- to post-treatment (depression, NA, PA, and mental health) significantly, except physical health, and the improvements were maintained until the follow-ups. The same thing occurred with the variables evaluated at pre-treatment and the 6- and 12-month follow-ups (well-being and general health status) because both variables improved significantly from pre to follow-ups. Hence, it is important to highlight that the PA intervention was not only able to decrease negative functioning measures significantly (PHQ-9 and PANAS-NA) but also to increase the positive functioning variables (PANAS-PA, quality of life, and well-being).

Our results showed that PP-based strategies focusing especially on the promotion of PA might have an impact on the decline in clinical symptomatology and the increase in positive functioning variables, which is consistent with other studies [46,47]. These results emphasize the benefits of including PP exercises and techniques in conventional treatments to promote positive emotions and positive functioning measures [19,20]. Recent studies suggest the importance of directly working on the promotion of positive emotions to improve depressive symptoms and achieve greater changes in positive emotion outcomes [48,49,50]. Nevertheless, the role of PP in interventions for depression has been poorly studied. The majority of the interventions for depression focus on reducing distress and negative emotions; however, it is has been recognized that well-being and the increase in positive emotions and behaviors contribute to mitigating negative emotions in life [51] and help to reduce stress reactivity [52,53]. Depression is characterized by high levels of NA. However, patients suffering from depression also have low PA, which, if not addressed, increases the consequences of the problem [11,54,55]. Moreover, PA has been found to contribute to better physical and psychological health and well-being [56], and so its improvement should be considered a core element of depression treatment. Our results show that depression can be addressed effectively not only by managing negative symptoms but also by increasing positive emotions [57,58].

Regarding the profile of patients who benefited more from this kind of intervention, sociodemographic variables and baseline scores on specific psychological measures were predictors of the changes in the primary and secondary outcomes. Overall, patients with lower income levels (i.e., under the minimum income level), lower educational levels (i.e., no education or only primary school), and worse scores on mental health and well-being before starting the intervention benefited more from the intervention. More specifically, *lower income levels* predicted greater improvements in depression severity at post-treatment (19.7% of variance), 6-month follow-up (10.6% of variance), and 12-month follow-up (28.3% of variance), as well as greater improvements in mental health and health status at the 6-month follow-up (11.0% and 21.6% of variance, respectively); *worse mental health* predicted greater improvements in well-being at the 6-month follow-up (7.9% of variance); *lower well-being* predicted greater improvements in depression severity at the 12-month follow-up (11.2% of variance); and finally, *lower education levels* predicted greater improvements in physical health at post-treatment (8.1% of variance), 6-month follow-up (16.0% of variance), and 12-month follow-up (19.6% of variance).

Regarding the education and income levels, our results pointed out that these variables were associated with intervention efficacy. In addition, we should highlight that this PA intervention was implemented in public PC contexts, where health status, education level, and income level are usually lower than in other psychological treatment settings. Thus, this study suggests that a brief self-guided IBI could help people in more disadvantaged situations. None of the other sociodemographic variables examined predicted treatment outcomes. This means that the PA intervention is equally useful for people of different ages, marital status, work status, and sexes, which supports its dissemination. This result is in line with previous studies that found no sociodemographic variables that moderated the results of IBIs for depression [59].

Regarding mental health and well-being status, our findings are congruent with the results obtained by an Individual Patient Data (IPD) meta-analysis of low-intensity interventions for depression [60]. This study found that higher severity of depressive symptoms at baseline was associated with a greater decrease after the completion of the treatment. Nevertheless, because the effect was relatively small, the authors remarked that it is more cautious to conclude that low-intensity interventions work similarly across all the ranges of depression severity. By contrast, another recent IPD meta-analysis found that baseline depressive symptom scores did not moderate treatment outcomes [59]. This finding does not agree with our results because it suggests that characteristics of the intervention (e.g., focused only on PA) or the participants (e.g., patients from public PC settings) could explain these differences. More research with larger samples and different IBIs is needed to draw firm conclusions.

Regarding the mechanism of change, the improvement in depression severity was the most common predictor of the change in quality of life (mental and physical), well-being, and health status. More specifically, greater improvements in *depression severity and NA at post-treatment* predicted greater improvements in physical health at post-treatment (8.2% and 5% of variance, respectively); greater improvements in *depression severity at the 6-month follow-up* predicted greater improvements in mental health and well-being at the 6-month follow-up (10.4% and 9.2% of variance, respectively) and well-being at the 12-month follow-up (11.1% of variance); greater improvements in *depression severity at the 12-month follow-up* predicted greater improvements in well-being and health status at the 12-month follow-up (23.9% and 8.2% of variance, respectively). Furthermore, the increase in PA was also a relevant variable in increasing well-being in the long term. Specifically, greater improvements in *PA at the 12-month follow-up* predicted greater improvements in well-being at the 12-month follow-up (12.8% of variance). Hence, our results point out two important mechanisms of change in well-being in the long term (12-month follow-up): the improvement in depressive symptoms (PHQ-9) and the improvement in PA. These findings suggest that even though our intervention is focused on PA, changes in well-being are explained by changes in both depressive symptoms and PA.

This study has some limitations. First, the sample was small, and so the results should be interpreted with caution. Future PP research needs to focus on increasing sample sizes [61]. Second, the predictions of the multiple regression analyses were based on the stepwise method, so the conclusion should be taken as preliminary (i.e., the stepwise method is based on a mathematical criterion and not a theoretical criterion). Third, in our sample, participants had mild or moderate depressive symptoms; thus, it would be important to investigate the effect of PP strategies on patients with more severe depression [22]. Finally, it was not possible to record the number of face-to-face sessions focused on treating depression with the GP (although the available information indicated that they were very scarce and brief) or the medication dose administered to each patient in the iTAU condition. Future studies should consider these variables and analyze whether participants in IBIs receive less medical attention, which could be a cost-effectiveness variable of this treatment. Future research could explore the efficacy of this PP intervention, but in a completely self-guided way, without the support of the GP [59].

Despite the exploratory nature of this work, we consider that this study has several strengths. It may be relevant to the field of PP interventions, as it provides preliminary data on the profile of patients who benefit most of them. It also identifies the mechanisms of change of the intervention showing the improvement in PA, and not only in depression, as a significant predictor of long-term change in well-being. Finally, the study was carried out in a real context in a naturalistic way, which informs us regarding the utility of low-intensity IBIs in PC settings. The findings of the present study suggest that PA promotion can be an effective approach to treat depression by helping to improve both negative symptoms and positive emotions [62]. Furthermore, the PP intervention can be delivered using the Internet and self-guided formats, thus responding to an essential challenge in the field of depression treatments: reaching everyone in need by using new and alternative ways to apply psychological interventions. Therefore, the use of digital interventions helps to fulfill this proposal by contributing to the accessibility and dissemination of evidence-based treatments in the PC context.

## 5. Conclusions

In conclusion, this was the first study to explore a low-intensity IBI based on PA promotion in the PC setting in Spain. The results suggest that PP strategies might have an impact on clinical symptomatology, but they also contribute to improvements in positive functioning measures. Moreover, income level, education level, mental health, and well-being were significant predictors of the changes in depression severity and positive functioning measures at the end of the treatment and in the follow-up period. In addition, two mechanisms were identified as being responsible for the change in long-term well-being: the improvement in depression severity and the improvement in PA. Nevertheless, these findings should be interpreted with caution given the exploratory nature of the analyses and the small sample size used. Finally, this study highlights the importance of implementing IBIs in PC to produce changes in both negative and positive dimensions of human psychological functioning and promote more comprehensive psychotherapy for depression [63]. Overall, this study provides further support for the application of PP techniques using an IBI [62,63,64]. Future research should take into account the profile and unique characteristics of each patient to provide reliable tools to guide the choice of effective treatments.

## Figures and Tables

**Figure 1 ijerph-17-08094-f001:**
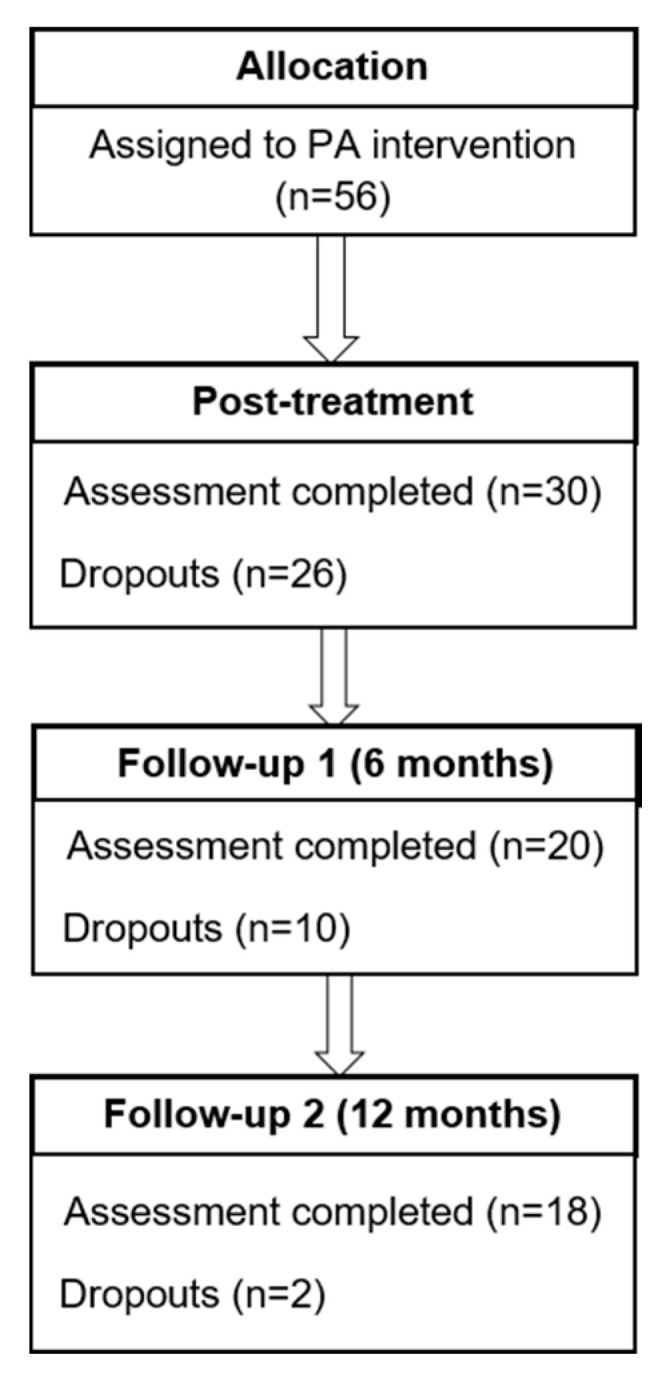
Patient flow diagram. PA: positive affect

**Table 1 ijerph-17-08094-t001:** Modules, objectives, and therapeutic content of Internet-Based Computerized Intervention (IBI).

Module	Objectives	Therapeutic Content
Learning to live	- Understanding the role of activity in mood regulation and our well-being. - Establishing and maintaining an adequate activity level and the relevance of choosing activities that are significant, with a personal meaning for the individual. - Learning the procedure to follow to schedule meaningful activities in daily life.	The role of activity in our well-being. Things we should do and things we can do: meaningful activities. The importance of daring, of getting involved with life. Seeking social support. Overcoming obstacles.
2. Learning to enjoy	- Learning about the effect of positive emotions in our lives. - Learning procedures to increase the likelihood of experiencing positive emotions, promoting the occurrence of pleasant activities to learn to enjoy the present moment.	Positive emotions, such as seeds or life anchors. Satisfaction with the present. Learning to generate good moments. The importance of smiling. Learn to identify, capture, and save good times.
3. Accepting to live	- Focusing on positive emotions related to the past (e.g., gratitude) or the future (such as optimism). - Identification and management of beliefs and behaviors that disturb the good moments.	Satisfaction with the past. Satisfaction with the future. Psychological well-being as a result of being active and practicing learned strategies.
4. Living and learning	- Understanding life as a continuous process of learning and personal growth. - Emphasizing the training in strategies to promote psychological strengths, resilience, and meaningful goals linked to important values.	Finding psychological well-being. Potential, talents, and life goals. Living with others, finding support in others. What do I want my future to be like?

**Table 2 ijerph-17-08094-t002:** Sociodemographic characteristics of the sample.

Sociodemographic Variables	%	n = 56
**Sex**		
Women	78.6%	44
Men	21.4%	12
**Age (years)**	44.14 (10.38) ^a^	
**Marital status**		
Single	16.1% ^b^	31
In a relationship/Married	55.4%	9
Divorced/Separated	16.1%	9
Widowed	1.8%	1
**Income level**		
Lower than the minimum income (<641,40€)	23.2%	13
Between 1–2 minimum incomes ^c^	25.0%	14
Between 2–4 minimum incomes	21.4%	12
**Work status**		
Student	3.6%	2
Housekeeper	3.6%	2
Subsidized unemployed	5.4%	3
Unemployed with no subsidy	12.5%	7
Employee	44.6%	25
Sick leave	8.9%	5
Retired	3.6%	2
Disability	3.6%	2
Others	3.6%	2
**Educational Level**		
No education	5.4%	3
Primary school	16.1%	9
Secondary school	33.9%	19
University studies	28.6%	16
**Living alone or with others**		
Alone	8.9%	5
Living with partner	16.1%	9
Living with partner and children	53.6%	30
Living with relatives	8.9%	5
Living with friends or neighbors	1.8%	1

Note. ^a^ These values are the mean and standard deviation of age, which is a continuous variable. ^b^ Due to missing values, the sum of the percentages of the categories of the “marital status”, “income level”, “work status”, “educational level”, or “living alone or with others” are not 100%. That is, we do not have the 30.4% of the information regarding “income level” (*n* = 17); the 10.7% of the information regarding “marital status”, “work status”, and “living alone or with others” (*n* = 6); and the 16.1% of the information regarding the “educational level (*n* = 9). ^c^ Between “1–2 or “2–4” minimum incomes means that the minimum income level is equivalent to “641,40€ (641,40€x 1)–1282,80€ (641,40€x 2)” or “1282,80€ (641,40€x 2)–2.565,60€ (641,40€x 4)”, respectively.

**Table 3 ijerph-17-08094-t003:** Descriptive statistics and repeated-measures ANOVA results of the primary and secondary outcomes.

	Pre*M (SD)*	Post*M (SD)*	FW6*M (SD)*	FW12*M (SD)*	*F*	Within-Group Effect Size, d [95% CI] Pre–Post	Within-Group Effect Size, d [95% CI] Pre-FW6	Within-Group Effect Size, d [95% CI] Pre-FW12
1. Depression Severity (PHQ-9)	15.79 (6.21)	10.57 (6.68)	8.63 (6.11)	9.75 (5.65)	*F*(2.51, 137.89) = 33.15, MSE = 20.32, *p* < 0.001	0.83 [0.55, 1.11]	1.14 [0.78, 1.49]	0.96 [0.61, 1.31]
2. Positive Affect (PANAS-PA)	17.63 (6.11)	20.75 (7.92)	22.27 (8.23)	22.95 (9.66)	*F*(2.37, 130.09) = 9.61, MSE = 41.42, *p* < 0.001	−0.50 [−0.78, −0.22]	−0.75 [−1.06, −0.43]	−0.86 [−1.15, −0.57]
3. Negative Affect (PANAS-NA)	26.82 (8.56)	22.23 (8.07)	21.21 (7.30)	22.02 (8.56)	*F*(2.54, 139.72) = 11.37, MSE = 37.47, *p* < 0.001	0.53 [0.28, 0.78]	0.65 [0.34, 0.96]	0.55 [0.25, 0.85]
4. Well-being (PHI)	4.21 (1.72)	-	5.70 (1.89)	5.59 (1.78)	*F*(2110) = 33.48, MSE = 1.15, *p* < 0.001	-	−0.85 [−1.12, −0.59]	−0.79 [−1.09, −0.49]
5. Mental Health (SF-12)	26.22 (8.08)	30.50 (10.43)	38.61 (10.83)	35.26 (13.42)	*F*(2.53, 139.31) = 20.65, MSE = 94.52, *p* < 0.001	−0.52 [−0.79, −0.26]	−1.51 [−1.98, −1.04]	−1.10 [−1.47, −0.74]
6. Physical Health (SF-12)	42.45 (9.83)	45.11 (11.32)	45.58 (10.15)	47.04 (10.66)	*F*(3, 165) = 4.66, MSE = 44.02, *p* = 0.004	−0.27 [−0.50, −0.03]	−0.31 [−0.59, −0.04]	−0.46 [−0.70, −0.22]
7. Health Status (EQ-5D)	47.86 (20.34)	-	62.68 (15.24)	66.57 (19.04)	*F*(1.65, 90.64) = 24.68, MSE = 268.55, *p* < 0.001	-	−0.72 [−1.00, −0.44]	−0.91 [−1.29, −0.53]

Notes. PHQ-9 = Patient Health Questionnaire-9; PANAS = The Positive and Negative Affect Scale; PHI = Pemberton Happiness Index; SF-12 = Short Form 12 Health Survey; EQ-5D = EuroQoL; PA = Positive affect; NA = Negative affect; Pre = Pre-treatment; Post = Post-treatment; FW6 = 6-month follow-up; FW12 = 12-month follow-up. MSE: Mean Squared Error.

**Table 4 ijerph-17-08094-t004:** Point-Biserial and Pearson’s correlations between proposed predictor variables.

Sociodemographic Variables and Pre-Treatment Scores	1	2	3	4	5	6	7	8	9	10	11	12	13	14
1. Sex	—													
2. Age	−0.13	—												
3. Marital status (binary variable)	−0.05	0.03	—											
4. Living alone or not (binary variable)	0.13	−0.04	0.29 *	—										
5. Educational level (binary variable)	−0.19	0.33 *	0.04	−0.04	—									
6. Work status (binary variable)	−0.17	0.03	0.02	−0.14	0.02	—								
7. Income level (binary variable)	−0.13	0.50 **	0.26	0.05	0.21	0.22	—							
8. Depression Severity (PHQ-9)	0.02	−0.09	-0.23	0.05	−0.20	−0.12	−0.24	—						
9. Positive Affect (PANAS-PA)	−0.03	0.04	0.07	0.08	0.08	0.15	0.25	−0.33 *	—					
10. Negative Affect (PANAS-NA)	−0.05	−0.20	0.04	0.26	−0.16	0.04	−0.14	0.55 ***	−0.33 *	—				
11. Well-being (PHI)	0.04	0.10	0.33 *	−0.15	0.13	0.15	0.23	−0.51 ***	0.50 ***	−0.40 **	—			
12. Mental Health (SF-12)	0.13	0.08	0.26	0.11	0.09	0.05	0.28	−0.43 **	0.50 ***	−0.21	0.53 ***	—		
13. Physical Health (SF-12)	−0.05	−0.30 *	0.15	0.15	0.23	0.22	0.08	−0.36 **	0.15	−0.25	0.09	−0.09	—	
14. Health Status (EQ-5D)	−0.03	−0.12	0.20	0.07	0.15	0.26	0.44 **	−0.43 **	0.36 **	−0.16	0.42 **	0.26	0.40 **	—

Notes. PHQ-9 = Patient Health Questionnaire-9; PANAS = The Positive and Negative Affect Scale; PHI = Pemberton Happiness Index; SF-12 = Short Form 12 Health Survey; EQ-5D = EuroQoL; PA = Positive affect; NA = Negative affect. Point-Biserial’s correlations were carried out among dichotomous and recoded binary variables and between dichotomous/recoded binary variables and continuous variables. Pearson’s correlations were carried out among continuous variables. * *p* < 0.05, ** *p* < 0.01, *** *p* < 0.001.

**Table 5 ijerph-17-08094-t005:** Models for sociodemographic variables and pre-treatment scores as predictors of change in primary and secondary outcomes.

Outcomes	Predictors	R	Adjusted R^2^	R^2^ Change	B	SE	β	t
Change in PHQ								
*Pre-Post*	Constant				−8.93	1.41		6.32 ***
	Level of incomes (binary variable)	0.47	0.20	0.22	5.57	1.73	0.47	3.22 **

*Pre-FW6*	Constant				−10.49	1.74		6.04 ***
	Level of incomes (binary variable)	0.36	0.11	0.13	4.99	2.13	0.36	2.35 *

*Pre-FW12*	Constant				−16.40	2.36		6.95 ***
	Level of incomes (binary variable)	0.55	0.28	0.30	6.57	1.83	0.47	3.60 **
	PHI	0.65	0.40	0.13	1.42	0.51	0.36	2.80 **

Change in PANAS positive								
*Pre-Post*	-	-	-	-	-	-	-	-
*Pre-FW6*	-	-	-	-	-	-	-	-
*Pre-FW12*	-	-	-	-	-	-	-	-

Change in PANAS negative								
*Pre-Post*	-	-	-	-	-	-	-	-
*Pre-FW6*	-	-	-	-	-	-	-	-
*Pre-FW12*	-	-	-	-	-	-	-	-

Change in PHI								
*Pre-FW6*	Constant				2.97	0.75		3.97 ***
	SF-12 (Mental Health)	0.32	0.08	0.10	−0.06	0.03	−0.32	2.07 *
*Pre-FW12*	-	-	-	-	-	-	-	-

Change in SF-12 (Mental Health)								
*Pre-Post*	-	-	-	-	-	-	-	-
*Pre-FW6*	Constant				19.23	3.51		5.48 ***
	Level of incomes (binary variable)	0.37	0.11	0.13	−10.26	4.29	−0.37	2.39 *
*Pre-FW12*	-	-	-	-	-	-	-	-

Change in SF-12 (Physical Health)								
*Pre-Post*	Constant				7.77	2.85		2.73 *
	Education level (binary variable)	0.32	0.08	0.11	−6.87	3.30	−0.32	2.08 *

*Pre-FW6*	Constant				10.58	3.010		3.52 **
	Education level (binary variable)	0.43	0.16	0.18	−10.01	3.490	−0.43	2.87 **

*Pre-FW12*	Constant				11.11	2.51		4.42 ***
	Education level (binary variable)	0.44	0.18	0.20	−8.76	2.91	−0.44	3.01 **

Change in EQ-5D								
*Pre-FW6*	Constant				26.30	4.40		5.98 ***
	Level of incomes (binary variable)	0.47	0.20	0.22	−17.21	5.39	−0.47	3.19 **
*Pre-FW12*	-	-	-	-	-	-	-	-


*Notes.* * *p* < 0.05; ** *p* < 0.01; *** *p* < 0.001. PHQ-9 = Patient Health Questionnaire-9; PANAS = The Positive and Negative Affect Scale; PHI = Pemberton Happiness Index; SF-12 = Short Form 12 Health Survey; EQ-5D = EuroQoL; PA = Positive affect; NA = Negative affect; Pre = Pre-treatment; Post = Post-treatment; FW6 = 6-month follow-up; FW12 = 12-month follow-up.

**Table 6 ijerph-17-08094-t006:** Models for changes in depression severity (PHQ-9) and affect (PANAS) as predictors of change in well-being (PHI), health-related quality of life (SF-12), and health status (EQ-5D).

Outcomes	Predictors	R	Adjusted R^2^	R^2^ Change	B	SE	β	t
Change in PHI								
*Pre-FW6*	Constant				1.06	0.25		4.33 ***
	Change in PHQ-9 *Pre-Post*	0.33	0.09	0.11	−0.08	0.03	−0.33	2.57 *

*Pre-FW12*	Constant				0.57	0.27		2.14 *
	Change in PHQ-9 *Pre-FW12*	0.50	0.24	0.25	−0.22	0.04	−0.88	5.84 ***
	Change in PANAS-PA *Pre-FW12*	0.63	0.37	0.14	0.08	0.02	0.37	3.73 ***
	Change in PHQ *Pre-FW6*	0.71	0.48	0.12	0.14	0.04	0.52	3.50 **

Change in SF-12 (Mental Health)								
*Pre-Post*	-	-	-	-	-	-	-	-
*Pre-FW6*	Constant				7.37	2.51		2.94 **
	Change in PHQ-9 *Pre-FW6*	0.35	0.10	0.12	−0.70	0.26	−0.35	2.72 **
*Pre-FW12*	-	-	-	-	-	-	-	-

Change in SF-12 (Physical Health)								
*Pre-Post*	Constant				−1.18	1.68		0.70
	Change in PHQ-9 *Pre-Post*	0.31	0.08	0.10	−0.44	0.21	−0.27	2.10 *
	Change in PANAS-NA *Pre-Post*	0.41	0.13	0.07	−0.34	0.17	−0.26	2.04 *
*Pre-FW6*	-							
*Pre-FW12*	-							

Change in EQ-5D								
*Pre-FW6*	-	-	-	-	-	-	-	-
*Pre-FW12*	Constant				11.56	4.39		2.64 *
	Change in PHQ-9 *Pre-FW12*	0.31	0.08	0.10	−1.19	0.49	−0.31	2.43 *

Notes. * *p* < 0.05; ** *p* < 0.01; *** *p* < 0.001. PHQ-9 = Patient Health Questionnaire-9; PANAS = The Positive and Negative Affect Scale; PHI = Pemberton Happiness Index; SF-12 = Short Form 12 Health Survey; EQ-5D = EuroQoL; PA = Positive affect; NA = Negative affect; Pre = Pre-treatment; Post = Post-treatment; FW6 = 6-month follow-up; FW12 = 12-month follow-up.

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
