# Peer review of "A Low-Intensity Internet-Based Intervention Focused on the Promotion of Positive Affect for the Treatment of Depression in Spanish Primary Care: Secondary Analysis of a Randomized Controlled Trial"

_ijerph, 2020, doi:10.3390/ijerph17218094_

Round 1

Reviewer 1 Report

The authors present research started on 2015-16, as a secondary analysis of the randomized control trial conducted by Gili et al. (2020), examining the role of an IBI focused on promoting PA in patients with depression in PC. I would like to be more positive towards the work, but there are certain errors in data analysis that do not allow me to be so. If published in the following way, it would be detrimental to the field and also for the reputation of the authors. I hope my contributions will be useful in modifying these analyses.

Major points:

The Data analysis: First, I was suprissed that authors have included a dichotomous variable such as sex in the correlation analysis, under Pearson coefficient. I find this alarming and it makes me doubt regarding other analyses. Not only that, they have included other ordinal variables for the correlations (as well as, other socio-demographic and dichotomous variables), which has a series of assumptions that show that are unknown to the authors.

Secondly, although it is possible to include a dichotomous variable as a predictor in a regression analysis, under very justified circumstances and with great care in its analysis, this is not possible for ordinal variables.

Authors also made conclusions on these analysis, see: "Moreover, income level, education level, mental health, and well-being were significant predictors of the changes in depression severity and positive functioning measures at the end of the treatment and in the follow-up period."

Other points:

-Authors need to explain if assumptions for ANOVA were reached. If so, auhtors need to include MSE in the ANOVAS

-It is difficult to understand the timing, which is essential for the research. Moreover the drop-outs. A consort flow diagram should be used for reporting an RCT

http://www.consort-statement.org

-As mentioned before, the prediction analysis is not posible on the sociodemographic predictors stated. Authors need to search professional advice on the data analysis, moreover on the method employed for this aim. In my opinion, stepwise multiple regression will not be the best approach to do so, but you need to review it with a professional and explain your decision in the manuscript.

-Most of the literature is almost from a decade ago, even if authors are using 61 references.

Minor points:

-t values are normally not reported in a negative way

Reviewer 2 Report

Overall interesting and challenging.

I have read through the paper with interest. I have a few suggestions to be addressed by the authors.

The abstract reads well and is informative, but in this section (line 42, page 1), a MDD definition (or just change for Major Depressive Disorder), would be desirable.

If available, this reviewer suggests adding additional identifiers (EudraCT number or ClinicalTrials.gov number) apart from the Current Controlled Trials registry (ISRCTN82388279), stated at lines 112 (page 3 of 25).

Perhaps table 1, sociodemographic characteristics of the sample (pages 3 and 4), would fit better at the beginning of the results section.

An exhaustive description of interventions (section 2.3. starting on page 5), and measures (section 2.4) might fit better as annexes (just summarizing interventions and measures in the Materials and Methods section).

The RCT reported should have been carried out in accordance with generally accepted ethical research standards that should be included as the last paragraph of the Materials and Methods section (line 274, page 9).

Reviewer 3 Report

1.       Please add some results in numbers in the abstract in addition to the summary of the results.

2.       Table 1: there is something wrong in presenting the numbers. In the second column, where you are supposed to put the mean (SD), you are actually presenting the percentage. So, please rename and clarify what is presented in each column.

3.       Table 1: Also, in the income level variable: the percentages presented are not correct, for example the “lower than the minimum income” should be 23.2% but in the table you put 33.3%... please check all again and correct them.

4.       Table 1: What do you mean by: “Between 1-2 minimum incomes” and “Between 2-4 minimum incomes”?

5.       In the discussion and conclusion, the authors need to highlight the results and indicate the strength and weaknesses of the study.

6.       Please check your reference list to make them consistent.

Round 2

Reviewer 1 Report

I am grateful for the great job that the authors have done. In particular, the inclusion of a flow chart and the revision of variables in the regression section. However, there are some limitations that drive me to the current decision.

The authors have changed parametric methods for non-parametric ones, for example, to do a correlation analysis. I am afraid that the assumptions of normality among others are not fulfilled in order to apply Pearson, but a Spearman method is not enough, and I doubt that it is even possible here:

E.g., in table 4, authors mix binary to binary, or binary and continuos correlations (but they report Pearson or Spearman correlations).
In that case, I would take out all these binary variables from the analysis and in some cases, considered the Point-Biserial Correlation. This is a formula developed to be equivalent both to Kendall'sτ and Spearman'sρ fro some specific cases that mightbe of . Please, see Cureton (1956) "Rank Biserial Correlation", Psychometrika, 21. This is one example, from the previous ones I have pointed out.

This kind of basic errors, even in second revision, make me doubt on the rest of the statistics. Thus, I am afraid I cannot accept the current manuscript.

Author Response

Thank you again for all your suggestions, as they have significantly improved our manuscript.

In SPSS, the Point-Biserial Correlation Coefficient is calculated just as the Pearson’s Bivariate Correlation Coefficient. That is, once the assumptions have been checked, you introduce the dichotomous (with 0 and 1) and the continuous variable and you click in Pearson correlation. This information is on the book of Field (2018): "We want to calculate a point-biserial correlation, and this is simplicity itself: it is a Pearson correlation when the dichotomous variable is coded with 0 for one category and 1 for the other (in practice, you can use any values because SPSS changes the lower one to 0 and the higher one to 1 when it does the calculations)." (pp. 536). This is also explained in several websites dedicated to statistics (e.g., https://www.statisticssolutions.com/point-biserial-correlation/; https://statistics.laerd.com/spss-tutorials/point-biserial-correlation-using-spss-statistics.php#procedure).

Following the reviewer’s and editor’s comment, we have used Point-Biserial’s correlations (instead of Spearman’s correlations) to analyse the relationships between the binary variables among them, as well as the relationships between the binary and continuous variables. As the reviewer can verify, the values of the correlations are the same as the first version of the manuscript, as the SPSS’s command to calculate these correlations is the same as Pearson’s correlations. Hence, there is no significant changes in these results.

Finally, we want to put on record that all statistical analyses have been supervised by a statistician and our analyses are trustworthy (i.e., we checked that the assumptions are met, all the analyses are explained and justified in the manuscript...).